# Development of pH-Sensitive Chitosan-*g*-poly(acrylamide-*co*-acrylic acid) Hydrogel for Controlled Drug Delivery of Tenofovir Disoproxil Fumarate

**DOI:** 10.3390/polym13203571

**Published:** 2021-10-16

**Authors:** Justin B. Safari, Alain M. Bapolisi, Rui W. M. Krause

**Affiliations:** 1Department of Chemistry, Faculty of Science, Rhodes University, Makhana 6140, South Africa; alainbapo@gmail.com; 2Department of Pharmacy, Faculty of Pharmaceutical Sciences and Public Health, Official University of Bukavu, Bukavu 570, Democratic Republic of the Congo; 3Center for Chemico- and Biomedicinal Research (CCBR), Faculty of Science, Rhodes University, Makhana 6140, South Africa

**Keywords:** smart drug delivery system, hydrogel, chitosan, tenofovir disoproxil fumarate, controlled delivery

## Abstract

The present study aimed to develop a pH-sensitive chitosan-based hydrogel for controlled delivery of an anti-hepatitis B drug, tenofovir disoproxil fumarate (TDF). Free radical polymerization was utilized to graft acrylamide and acrylic acid using *N*,*N*-methylene bisacrylamide as the crosslinker. Physicochemical characterization confirmed the synthesis of thermally stable chitosan-g-poly(acrylamide-co-acrylic acid) hydrogels with well-defined pores within a fibrous surface. The prepared hydrogels exhibited pH and ionic strength sensitivity, with the swelling significantly lower under acidic and strong ionic strength conditions but higher in neutral and basic solutions. In addition, cytotoxicity studies on HeLa cell lines proved the cytocompatibility of the drug delivery material and its readiness for physiological applications. The encapsulation of TDF in the hydrogels was optimized and an encapsulation efficiency and a drug loading percentage of 96% and 10% were achieved, respectively. More interestingly, in vitro release studies demonstrated a pH-dependent release of TDF from hydrogels. The release at pH 7.4 was found to be up to five times higher than at pH 1.2 within 96 h. This further suggested that the newly developed hydrogel-loaded TDF could be proposed as a smart delivery system for oral delivery of anti-hepatitis B drugs.

## 1. Introduction

Hepatitis B virus (HBV) infections remain a significant global public health problem [1]. In 2017, the World Health Organization (WHO) reported that 257 million people suffer from chronic HBV infection, and nearly 887,000 deaths were registered worldwide due to the hepatitis B virus pathogen [2]. In general, there are two different strategies used in the treatment of HBV infection: the short-term therapy based on immune-modulators, such as standard or PEGylated interferon-α; and the long-term regime composed of the nucleosides and nucleotides analogues, such as tenofovir, lamivudine, entecavir, telbivudine, etc. [3]. The main goal of the therapeutic approach to treat HBV infection is to ensure a cure without relapse, thereby preventing death or the development of hepatocellular carcinoma. Furthermore, it is desirable to prevent transmission and the emergence of drug resistance during treatment. Therefore, the therapeutic actions of HBV drugs commonly employed are expected to lower the viral load to undetectable levels, decrease the disease rate, and reduce the rate of evolution of drug-resistant HBV [4]. However, the lengthy medication period of HBV infection leads to poor patient compliance that promotes the development of drug resistance. This significantly affects the efficacy of existing drugs and underlines the need for long-acting formulations that control and sustain the release and could thereby improve the management of HBV infections.

In recent decades, intelligent or smart drug delivery systems (SDDSs) capable of controlling the location, the time, and/or the rate of drug release by the action of external stimuli have captured many researchers’ attention. This results from several advantages to achieve therapeutic results, such as their potential to increase drug efficiency, reduce toxic side effects, enhance drug absorption, facilitate access to the target site, and control the drug input within the required time [5]. In a particular way, polymers play an important role in the design of SDDSs; they are widely used to develop various types of delivery systems such as micelles, particulate systems (beads, micro-and nanoparticles), hydrogels, etc. [6].

Hydrogels are three-dimension polymeric networks based on crosslinked hydrophilic polymers that retain a considerable amount of water or biological fluid without dissolving while maintaining their 3D structure [7,8]. Depending on the interactions within the crosslinked structure, hydrogels can be classified as physically crosslinked (when non-covalent interactions predominate), chemically crosslinked (when they are formed by covalent bonds), or mixed (i.e., containing both interactions). The constituting polymers can also be grouped as synthetic, natural, or hybrid [9]. In general, hydrogels exhibit good biocompatibility, biodegradability, low toxicity, and tunable physical properties (such as surface characteristics, swelling, stimuli-responsiveness, and mechanical strength), making them promising material for controlled drug delivery [10]. To achieve these properties, biomaterials (such as chitosan and its derivatives) have been extensively studied in hydrogel formation, and have proven to be good materials for delivering various drugs [11,12,13]. In addition, due to their Hamaker constant similar to the one of water, hydrogels are characterized by a high aqueous dispersion in biological fluids than other drug delivery materials [14]. Thermosensitive chitosan-based hydrogels containing microsphere-loaded tenofovir for vaginal drug delivery were recently reported by Yang et al. [15]. However, the use of pH-sensitive chitosan-based hydrogels for the delivery of tenofovir and targeting oral delivery has barely been investigated.

Based on the above considerations, the goal of this study was to develop a pH-sensitive hydrogel based on chitosan grafted with acrylic acid and acrylamide using free radical polymerization. This was carried out to evaluate the loading and release capacity for tenofovir disoproxil fumarate (TDF). The physicochemical properties of the hydrogel alone and the TDF-loaded hydrogel were assessed, as was the cytocompatibility.

## 2. Materials and Methods

### 2.1. Chemicals

Chitosan low molecular weight (CS), acrylic acid (AA), *N*,*N* methylene bisacrylamide (MBA), Mono-potassium phosphate (KH_2_PO_4_), and sodium phosphate dibasic (Na_2_HPO_4_) were purchased from Sigma-Aldrich (Darmstadt, Germany). Tenofovir disoproxil fumarate (TDF) was donated by Aspen Pharmacare (Durban, South Africa). Acrylamide (AAm), hydrogen chloride (HCl), and HPLC grade acetonitrile were purchased from Merck (Darmstadt, Germany). Ammonium persulfate and orthophosphoric acid were purchased from Saarchem (Wadeville, South Africa). Glacial acetic acid, sodium hydroxide (NaOH), magnesium chloride (MgCl_2_), sodium chloride (NaCl), potassium chloride (KCl), methanol, and acetone were purchased from Minema (Roodepoort, South Africa). Almunium chloride (AlCl_3_) was purchased from Fluka Chemika (Buchs, Switzerland). Ultra-pure water was HPLC grade 18 mega Ohm water, prepared using a Milli-Q academic A10 water purification system (Millipore^®^ Bedford, MA, USA).

### 2.2. Methods

#### 2.2.1. Synthesis of Chitosan Grafted Poly(acrylamide-*co*-acrylic acid)

A free-radical polymerization method was used to synthesize the chitosan-based hydrogel; the procedure described by Mahdavinia et al. was followed with slight modifications regarding reagents amounts and reaction conditions [16]. In a 250 mL two-neck reactor, 500 mg of chitosan was dissolved in 50 mL of acetic acid solution (1% solution). The reactor was placed in an oil bath on a hot plate equipped with a magnetic stirrer pre-set at 60 °C and allowed to stir (600 rpm) for 15 min under nitrogen atmosphere. Afterward, 100 mg of ammonium persulfate (5 mL of 2% APS solution) were added to the chitosan solution and allowed to stir for an additional 10 min. Subsequently, 500 mg of acrylic acid (5 mL of 10% AA solution) and 1500 mg of acrylamide (15 mL of 10% solution of AAm) were simultaneously added to the reaction mixture; hence, the ratio chitosan/synthetic monomer was 0.25. Following monomer addition, variable amounts of *N*,*N* methylene bisacrylamide 100–200 mg (5–10 mL of 2% MBA solution) (Table 1) were added, and the reaction mixture was continuously stirred for 60 min at 60 °C. At completion, the product was cooled at room temperature and then neutralized to pH 8 by a solution of NaOH (1 M). Then, about 250 mL of methanol was added to the gelled product while stirring for 24 h at room temperature for maximum dewatering. The solvent was filtered out and the gel particles were rinsed with fresh methanol (2 × 50 mL) and dried at 50 °C in an oven.

The prepared chitosan grafted Poly(acrylamide-*co*-acrylic acid) was then saponified in 50 mL of an aqueous solution of NaOH (1 M) in a loosely stopper 100 mL flask at 100 °C for 60 min. The reaction product was cooled to ambient temperature and was adjusted to pH 8 by adding a solution of acetic acid 10%. The dewatering and drying procedures were followed again as described in the previous paragraph.

#### 2.2.2. Swelling Studies

The filtration method was used to estimate the swelling capacity of the synthesized hydrogel. Briefly, a mass between 10 mg and 50 mg of the hydrogel was accurately weighed. The initial mass was recorded as *W*_0_ and placed in a beaker, then an excess of swelling medium (50–100 mL) was added and left in contact with the hydrogel for two hours. A piece of filter paper pre-saturated with distilled water was weighed; its mass was recorded as *W*_1_ and was placed in a Büchner funnel. The swollen sample was filtered under vacuum to remove any excess fluid from the gel. The filter paper (with the swollen gel on it) was taken out and weighed again, and its final mass was recorded as *W*_2_. The swelling ratio (*SR*) was calculated using Equation (1) [17]. This was repeated at least three times.
(1)$$SR=\frac{W_{2}-W_{1}-W_{0}}{W_{0}}$$

To study the pH sensitivity of the hydrogel, the swelling ratio was studied at different pH values in buffer solutions: pH 1.2 (HCl/KCl), pH 4 (acetate buffer), pH 5.8 (phosphate buffer), pH 7.4 (phosphate buffer), and pH 9.8 (carbonate buffer). A similar procedure was used to assess the ionic strength sensitivity of hydrogel using solutions of NaCl, MgCl_2_, AlCl_3_ at different concentrations (0.05, 0.1, and 0.15 M).

#### 2.2.3. Drug Encapsulation and In Vitro Release

##### Encapsulation of TDF

Tenofovir disoproxil fumarate (TDF) was chosen as an anti-hepatitis B drug model to investigate the loading and release properties of the synthesized hydrogel. The encapsulation was carried out using the swelling equilibrium method [18], where a defined amount of dried hydrogel (100 mg) was soaked in a fixed volume (50 mL) of a solution of TDF at room temperature. After 24 h, the swollen hydrogel was removed from the solution, washed with HPLC grade water to remove residual TDF, and then dried in the oven at 40 °C. The concentration of TDF was determined before and after soaking using a validated HPLC method (Data not shown). An Agilent 1100 Liquid Chromatography series equipped with a quaternary pump (G1311A), degasser (G1322A), diode array detector (G1315B), and manual injector (G1328B) was used for HPLC analysis with a Luna^®®^ LC column (5 μm C18, 100 Å, 250 × 4.6 mm i.d.). The mobile phase was a mixture of acetonitrile and water (65:35), eluted in isocratic mode at a flow rate of 1 mL/min. The sample injection volume was 20 µL, and the detection wavelength was 259 nm.

The encapsulation efficiency (EE) and the drug loading (DL) were calculated using the formulas (2) and (3), respectively. The EE and DL were optimized using four different concentrations of TDF, namely 100, 150, 200, and 250 µg/mL.
(2)$$EE\left(wt.\%\right)=\frac{W_{0}-W_{f}}{W_{0}}\times 100\;$$
(3)$$DL\;\left(wt.\%\right)=\frac{W_{0}-W_{f}}{W_{dH}}$$
where *W*_0_ and *W_f_* are the total weight of TDF in the solution before and after soaking the dried hydrogel in the solution of TDF, respectively, and *W_dH_* is the weight of the dried hydrogel. All the encapsulation experiments were performed in triplicate.

##### In Vitro Release of TDF

The concentration with optimum EE and DL of TDF, being 200 µg/mL, was considered for in vitro release studies. In a dialysis bag (dialysis tubing cellulose membrane, flat width 25 mm (1.0 in), Sigma-Aldrich), dried TDF-loaded hydrogel was immersed in 25 mL of buffer solutions (pH = 1.2 and 7.4, I = 0.1 M). The hydrogel samples were submerged at 37 °C for 96 h while maintaining a constant shaking (100 rpm), and, at selected time intervals (0.5, 1, 1.5, 2, 4, 8, 12, 24, 48, 72 and 96 h), an aliquot (5 mL) of the release milieu was withdrawn and immediately substituted with 5 mL of the fresh buffer solution. The amount of TDF released was quantified by HPLC for each aliquot, and the cumulative percentage of TDF release was calculated using Equation (4).
(4)$$\mathrm{Cumulative}\;\mathrm{drug}\;\mathrm{release}\;=\frac{Ve\sum_{i}^{n-1}Ci+V_{0}Cn}{m}\times 100$$
where m represents the amount of TDF in the hydrogel, *V*_0_ is the volume of the release medium (*V*_0_ = 25 mL), and *Cn* is the concentration of TDF in the nth sample. All TDF drug releases were performed in triplicates.

#### 2.2.4. Characterization

##### Fourier Transform Infra-Red Spectroscopy (FTIR)

A PerkinElmer Spectrum 100 FT-IR Spectrometer was used for the FT-IR analysis. The IR spectra were obtained by the attenuated total reflection (ATR) method. For each experiment, 16 scans were performed in the frequency range from 650 to 4000 cm^−1^, and the software Origin Pro 9.0 (OriginLab Corporation, 2012) was used to process and plot the data. The signals from the functional groups of CS-P (AAm-AA) loaded TDF (CS-P (AAm-AA)-TDF) were compared to those of empty hydrogel, chitosan, and TDF.

##### Powder X-ray Diffraction Spectroscopy (XRD)

XRD was used to compare the crystalline nature of CS, CS-P (AAm-AA), CS-P (AAm-AA)-TDF, and TDF. Analyses were performed using a nickel filter, and Cu-Ka radiation at 1.5404 Angstrom, and the scans were run at the 2–θ range from 10 to 100° with a slit width of 6.0 mm at a scanning speed of 1° min^−1^.

##### Thermal Analysis

Thermogravimetric analysis (TGA) and differential scanning calorimetry (DSC) were used to study the thermal behavior of CS compared to CS-P (AAm-AA) hydrogels. A Perkin TGA 4000 was used to study the thermal stability. A sample of 2–5 mg of the material was heated at a heating rate of 10 °C per minutes from 30 to 600 °C in an inert nitrogen atmosphere (20 mL/min).

A PerkinElmer DSC 6000 was used to assess the change in heat flow. The sample (3–5 mg) was placed into an aluminum pan and heated under a nitrogen gas atmosphere (20 mL/min) from 30 °C to 350 °C at a heating rate of 10 °C per minute. An empty aluminum pan was used as a reference.

##### Scanning Electron Microscopy (SEM) and Energy-Dispersive X-ray Spectroscopy (EDS)

The SEM and EDS were carried out to analyze the morphology and elemental composition of the synthesized hydrogel by using an INCA PENTA FET coupled to VAGA TESCAM energy dispersive X-ray spectroscopy. The hydrogel was swelled to equilibrium (24 h) in HPLC-grade water at room temperature and subsequently lyophilized in a freeze-dryer using a Lyo Lab 3000 lyophilizer Apollo Scientific CC. The lyophilized hydrogel was fixed on aluminum stubs, coated with gold, and then observed with SEM. The same was completed for cytotoxicity studies.

##### Cytotoxicity Assays

To evaluate the cytotoxicity of the synthesized drug carrier, the hydrogels (CS-P (AAm-AA) were incubated at a fixed concentration of 50 µg/mL in 96-well plates containing HeLa cells for 24 h. The number of cells surviving to hydrogel exposure was determined using the resazurin-based reagent and reading resorufin fluorescence in a multi-well plate reader. Results were expressed as percentage cell viability, based on fluorescence reading in treated wells versus untreated control well. Emetine (which induces cell apoptosis) was used as a positive control drug standard.

## 3. Results and Discussion

### 3.1. Synthesis of Chitosan Grafted Poly(acrylamide-co-acrylic acid)

Figure 1 shows the reaction scheme for the synthesis of chitosan grafted poly(acrylamide-*co*-acrylic acid) (CS-P (Aam-AA). Upon heating, ammonium persulfate used as an initiator generates sulfate radicals which are very powerful oxidants [19]. The generated radicals then reacted with -OH groups of chitosan to create the free radical site on the backbone of CS to initiate polymerization reaction of chitosan with acrylamide and acrylic acid. The addition of AAm and AA result in simultaneous polymerization, and MBA creates crosslinking between the different chains [11,20]. The last stage of the synthesis, which consisted of treating the hydrogel with sodium hydroxide, resulted in the transformation of the amide function into carboxylate followed by the saponification of the carboxylic group [16].

#### 3.1.1. Thermal Gravimetric Analysis

TGA was performed to assess the changes in thermal properties after grafting AAm and AA into chitosan and the creation of a crosslinked structure. The TGA thermogram of CS (Figure 2A) shows that its decomposition could be split into three successive zones of weight loss. The first one goes from 30–230 °C, representing a loss of 11.1%, and can be attributed to the loss of the water and volatiles from the sample. The second zone exhibits a sharp decomposition from 230–385 °C (42.9%); this sudden fall in weight can be due to the fragmentation of the backbone of chitosan. The last zone goes from 385–600 °C (20.5%); this last gradual decomposition can be attributed to the complete degradation of the polymer. However, the TGA thermogram of CS-P (AAm-AA) (Figure 2A) showed a weight loss across four distinct zones: the first one, 30–190 °C (7.1% weight loss), can again be attributed to the loss of moisture and volatiles; and the second, 190–321 °C (20.7%), and the third, 321–500 (29.4%), are attributed to the loss of grafted chains and fragmentation of the backbone, respectively. The fourth zone, 500–600 °C, represents the complete degradation of the material, similar to the previous non-grafted sample [21]. The difference between the TGA thermograms of CS and CS-P (AAm-AA), which are depicted in the graph of the first-order derivatives (Figure 2B), confirm the crosslinking of CS and the grafting of AAm and AA, indicating an improvement in the thermal stability of CS.

#### 3.1.2. Differential Scanning Calorimetry (DSC)

The DSC thermograms of chitosan presented in Figure 3 show two important thermodynamics events. Firstly, an endothermic peak between 30 °C and 119 °C presents a glass transition temperature (Tg) at 57 °C and enthalpy (∆H) of 156 J/g which might be associated with the loss of bound water and the transition from hard material to a soft and rubbery state. Secondly, an exothermic peak ranged between 280 and 333 °C with a Tg of 305 °C and an ∆H of −128 J/g, possibly indicating the degradation of chitosan [11], consistent with the above TGA weight loss at this temperature. Whereas the CS-P (AAm-AA) hydrogel thermogram (Figure 3) shows two curves: an endothermic one between 25 °C and 132 °C with a Tg at 65 °C and with ∆H of 211 J/g, and an exothermic peak between 273 °C and 305 °C with a Tg at 290 °C and with ∆H of −12 J/g. The shifts in both glass transition temperatures and enthalpies between the starting polymer and the hydrogel could be due to the improved stability of the gel network after crosslinking [22].

#### 3.1.3. Scanning Electron Microscopy (SEM) and Energy-Dispersive X-ray Spectroscopy (EDS)

Freeze-dried samples of CS-P (AAm-AA) hydrogel were analyzed to study their surface morphology and composition by SEM-EDS. To corroborate the hydrogel structure, elementary analysis using EDS was carried out (Figure 4C). This confirmed the presence of C, O, N, and Na. The surface morphology of hydrogel is a critical parameter in the design of hydrogels for drug delivery application since it affects the swelling characteristics and drug release behavior [23]. As shown in the micrographs (Figure 4A,B), CS-g-P(AAm-AA) hydrogels could be ideal as hydrogel drug carriers, as the surface of the freeze-dried hydrogels are rough and fibrous with numerous pores resulting from a good swelling as intrinsic properties of the synthesized hydrogels. However, the swelling capacity can be affected by extrinsic factors, such as pH or ionic strength. We therefore investigated the effects of the two external factors on the swelling of CS-*g*-P(AAm-AA).

### 3.2. Swelling Studies and Cytocompatibility

#### 3.2.1. pH-Sensitivity of CS-P (AAm-AA) Hydrogel

Figure 5 shows changes in the swelling behavior of CS-P (AAm-AA) hydrogel at different pH values. At pH 1.2 and 4, the swelling ratio (SR) was very low, while it was found to increase at pH 5.8 and 7.4. Yet, at pH 9.8, the SR started to decrease even though the value was still relatively high (Figure 5). At low pH (1.2 and 4), the swelling capacity was inadequate because the pH is below the pKa (4.75) of the carboxylic acid; hence, the function was protonated, but at high pH (5.8, 7.4 and 9.8), the swelling capacity increased because the pH was above the pKa of carboxylic acid. In that condition, a considerable number of –COOH functions would have been deprotonated and the electrostatic repulsion between the –COO- increased the swelling capacity. Similarly, the –COO- group is more hydrated than the –COOH group [24].

#### 3.2.2. Ionic Strength Sensitivity of (AAm-AA) Hydrogel

The swelling behavior of the hydrogel was studied in solutions of different salts (NaCl, MgCl_2,_ and AlCl_3_) and at various concentrations (0.05, 0.1, and 0.15 M). The result presented in Figure 6 revealed that, in most cases, the swelling capacity of the CS-P (AAm-AA) hydrogel decreases when the concentration of the salt increases (Figure 6A–C). This means that the change in ionic strength of the solution also affects the swelling behavior of the material. On the other hand, it shows that, as the valence of the metal in the salt increases, the swelling capacity is lowered. The decrease in the swelling capacity may be explained by the shielding effect created by the presence of a cation in the solution. The negatively charged groups (-COO-) presented in the chain of the hydrogel are surrounded by the cations which decrease the hydration [11]. Interestingly, for the concentration of 0.15 M (Figure 6D), similar to the physiological ionic strength, the sodium chloride demonstrated better swelling of the hydrogels compared to its counterparts. This suggests that the developed drug carrier could be applied for oral or intravenous delivery.

#### 3.2.3. Evaluation of Cytotoxicity of CS-P (AAm-AA)

Cytocompatibility is a relevant factor to consider when choosing a material for drug delivery application [20]. Although the non-cytotoxicity, biodegradability, and biocompatibility of chitosan have been demonstrated by multiple studies, its derivatives have to be carefully assessed for cytocompatibility before further biomedical application [25]. Hence, a study on cytocompatibility of the synthesized CS-P (AAm-AA) hydrogel was performed by studying cell viability of HeLa cells in the presence of the hydrogel using resazurin-based reagent. As depicted in Figure 7, all the hydrogel formulations were found to be non-cytotoxic at 50 µg/mL. Cytotoxicity of the hydrogel is also related to the amount of the crosslinker (*N*,*N* methylene bisacrylamide); it is observed that the higher the amount of the crosslinking agent, the lower cytotoxicity in the hydrogel. This profile is similar to the swelling profile of F1, F2 and F3; when the swelling is higher, the hydrogel is more dispersed in the cell culture medium and, hence, activity of the hydrogel will be more efficient. This result suggests that all the prepared hydrogels have a good cytocompatibility and that the amount of the crosslinker affects positively the cytotoxicity.

Overall, the proven cytocompatibility, thermal stability and good surface morphology with well-defined pores and optimum swelling capacity in physiological conditions indicate that the synthesized hydrogel formulations, CS-P (AAm-AA), could be suitable for controlled drug delivery applications.

### 3.3. Drug Encapsulation and In Vitro Release

#### 3.3.1. Optimization of the Drug Encapsulation

The chosen drug model TDF is a prodrug of tenofovir, a potent nucleotide analogue, that acts by inhibiting the reverse-transcriptase, due to its high activity against hepatitis B virus. TDF is used as a first-line drug in the treatment of HBV infection, but it does suffer from some slight solubility issues [26]. In this section, we investigated the effect of TDF concentration on the percentage of drug loading (DL) and the encapsulation efficiency (EE) to know the suitable concentration of the drug that must be used for further release studies. Four concentrations (100, 150, 200, and 250 µg/mL) were selected for this optimization. As shown in Figure 8, the EE was the highest at 100 µg/mL with a gradual decrease as the drug concentration increased. On the other hand, the DL increases with the increase in drug concentration. This contradicts the results of Hanna and Saad (2019), who found that the % DL and EE both increased with an increase in the concentration of the drug when encapsulating ciprofloxacin in xanthan gum-chitosan based hydrogel [27]. This may be because ciprofloxacin is highly soluble in water, unlike TDF.

In addition, the concentration of the crosslinking agent affects both the EE and DL. Formulations F1, F2 and F3 containing 10, 15 and 20 mg of MBA, respectively, show the same tendencies at all concentrations; the lower the crosslinking agent is, the higher both EE and DL are. At the concentration of 200 µg/mL, both the values of EE and DL were relatively high in all formulations compared to other concentrations. It was observed that the DL of F1, F2 and F3 are 10.29 ± 0.09, 9.72 ± 0.09 and 8/64 ± 0.14%, respectively, while the EE are 96.41 ± 0.29, 91.27 ± 0.89 and 81.15 ± 1.34, respectively. Therefore, the hydrogels formulations loaded with 200 µg/mL solution of TDF were selected for further characterization and release studies.

#### 3.3.2. FTIR Spectroscopy

FTIR was used to assess the functional groups present on chitosan and to see how they are affected following various modifications. The spectral analysis (Figure 9) of pure chitosan indicates the presence of its characteristic vibration peaks. The broadband in the region of 3355–3286 cm^−1^ corresponds to -NH_2_ and -OH stretching vibrations. Absorptions bands at around 2869, 1641 and 1020 cm^−1^ are attributed to C-H, C = O and C-O-C stretches, respectively, while the vibration at 1584 is the -NH- bend [28]. The CS-P (AAm-AA) spectrum shows that the peaks for carboxylic acid and amide bonds from AA and AAm cannot be distinguished because they overlap with the characteristic peaks of pure CS. Hence, the successful grafting of AAm and AA into the backbone of CS can only be confirmed by the noticeable increase in the intensity of the C=O stretches as well as its shift to higher frequency at 1657 cm^−1^, which suggests a high predominance of C=O groups in CS-P (AAm-AA) compared to pure CS. On the other hand, the saponification of CS-P (AAm-AA) resulted in the formation of -COONa groups whose presence is confirmed by the increased intensity of the peak at 1541 cm^−1^ in the spectrum of CS-P (AAm-AA). The disappearance of TDF characteristic peaks and the absence of noticeable changes in the spectrum of the TDF-loaded hydrogel (CS-P (AAm-AA)-TDF), when compared with CS-P (AAm-AA), are likely due to the low amount of TDF as well as the entrapment of the drug within the hydrogel [27].

#### 3.3.3. Powder X-ray Diffraction Spectroscopy

Powder X-ray diffraction was run to assess the change in the structure of chitosan after the chemical modification with AAm and AA and to see how encapsulation of TDF affects the crystalline structure of the material. Figure 10 presents the XRD pattern of CS, CS-P (AAm-AA), CS-P (AAm-AA)-TDF and TDF hydrogel. For chitosan, the spectrum shows one broad and weak peak at 2θ = 20°, indicating its semi-crystalline nature [11]. In the spectrum of CS-P (AAm-AA), the peak at 2θ = 20° is still present but with a weaker intensity. The decrease in intensity suggests a modification in the structure of chitosan and the reduction in its ability to form hydrogen bonds [29,30]. The TDF powder pattern shows a typical crystalline nature [31]. However, after encapsulation of TDF in the hydrogel, the material presents only a very broadband, which suggests the amorphous CS-P (AAm-AA)-TDF profile.

#### 3.3.4. In Vitro Release of TDF

In the design of a drug delivery system, choosing a drug model is an important stage. At this stage, the stability of the drug after encapsulation in the carrier is relevant. Interaction of drug encapsulated with the carrier and the solvent may cause denaturation of the drug which can affect its structural stability and biological activity [11]. UV-Visible spectroscopy was used to assess the chemical integrity of TDF before encapsulation and after release. As depicted in Figure 11, there is no change in λ_max_ (259 nm) of TDF before encapsulation and after release from the hydrogel; this result suggests that the encapsulation of TDF within the hydrogel does not change the integrity of the structure.

Many variables influence in vitro drug release from hydrogels; these include its composition, swelling behavior, drug affinity for the polymer gel matrix (amount of drug content) and the release medium [32]. In this study, the release experiments of Tenofovir disoproxil fumarate from TDF-loaded hydrogel were carried out in buffer solution at pH 1.2 and 7.4, which are physiological pHs of stomach and intestine, respectively. As shown in Figure 12A,B, at both pHs, all the formulations present a dual release profile, namely initial burst release (first 12 h) followed by a sustained release for 96 h. The formulation F3 made of 200 mg of *N*,*N* methylene bisacrylamide remarkably yielded to a high cumulative drug release. After 96 h, formulations F1, F2 and F3 released only about 5%, 6% and 10%, respectively, at pH 1.2. Whereas, at pH 7.4 about 39%, 39% and 53% were released from F1, F2 and F3, respectively. The rate of release is high in neutral pH compared to acid pH, which is ideal for slow release in the intestine. This behavior is likely due to the hydrogel swelling more in neutral media than in acid media. In acidic conditions, there is the formation of hydrogen bonds between COOH and OH groups that retard the release of the drug. This type of hydrogel was reported in similar systems in the literature [33,34]. Moreover, we observe that only a tiny quantity of the drug was released in acidic media; this feature may be beneficial for long-term oral administration that makes use of the pH differential between the stomach and the intestine, as it has been reported for other drug delivery systems [35].

## 4. Conclusions

In this study, pH-sensitive chitosan grafted poly(acrylamide-*co*-acrylic acid) hydrogel was prepared by free radical polymerization reaction and characterized by FTIR, DSC, TGA, XRD, EDS and SEM techniques. The prepared hydrogel was found to be non-cytotoxic at 50 µg/mL. The swelling ratio was mainly dependent on the pH and ionic strength, with maximum swelling at neutral or basic conditions and lower salt concentrations. Optimum conditions for encapsulation and release of TDF were investigated. The release rate was a function of pH and hydrogel composition, with the maximum release at pH 7.4 being five times higher than at pH 1.2 after 96 h for the formulation made with the highest amount of cross-linker (200 mg of *N*,*N*-methylene bisacrylamide). The findings suggest that the developed drug delivery systems have the potential to be applied for controlled delivery of hepatitis B drug, such as TDF, potentially for oral routes. However, further in and ex vivo studies are necessary to demonstrate the pharmacological activity as well as the pharmacokinetic profile of the developed formulation.

## Figures and Tables

**Figure 1 polymers-13-03571-f001:**
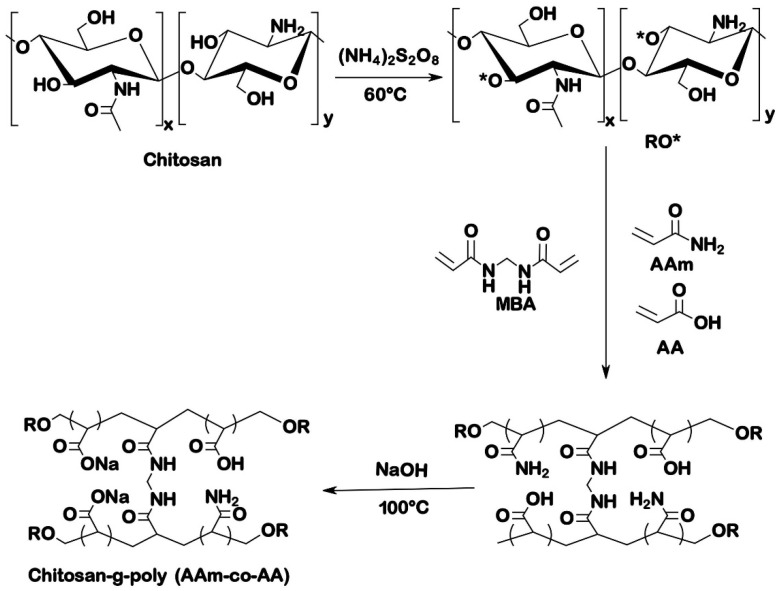
Reaction scheme for the synthesis CS-g-poly (AAm-AA) hydrogel.

**Figure 2 polymers-13-03571-f002:**
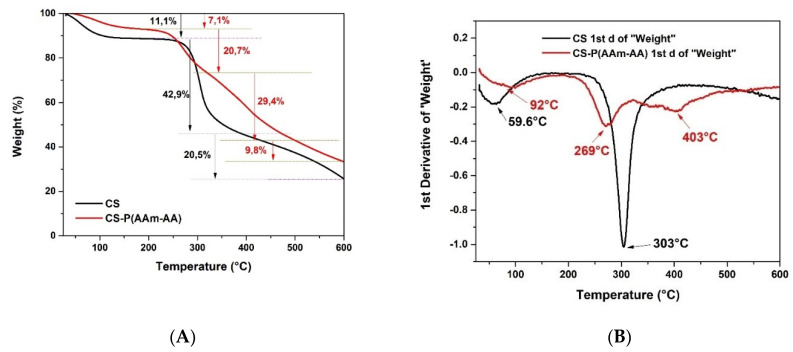
TGA thermogram of CS and CS-P (AAm-AA) (**A**) graph of the first-order derivatives of the weight loss of CS and CS-P (AAm-AA) (**B**).

**Figure 3 polymers-13-03571-f003:**
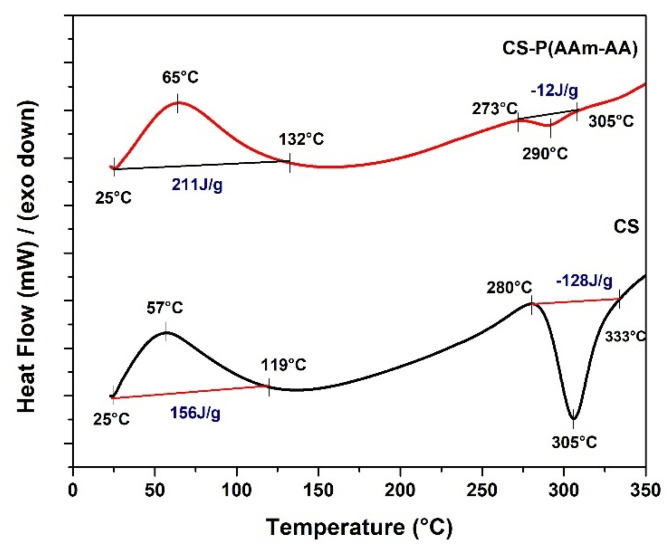
DSC thermogram of CS and CS-P (AAm-AA).

**Figure 4 polymers-13-03571-f004:**
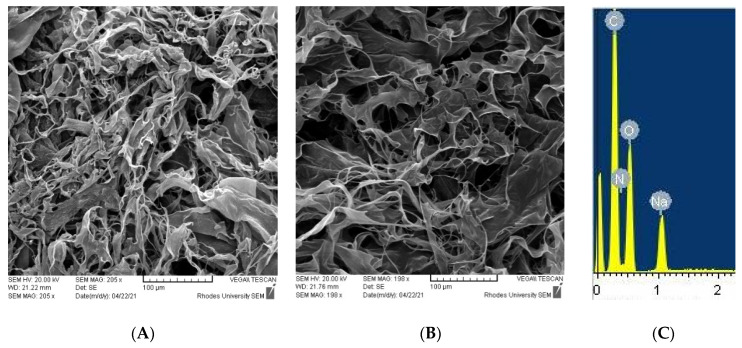
(**A**,**B**) SEM microphotographs of CS-P (AAm-AA), (**C**) EDS spectrum of CS-P (AAm-AA).

**Figure 5 polymers-13-03571-f005:**
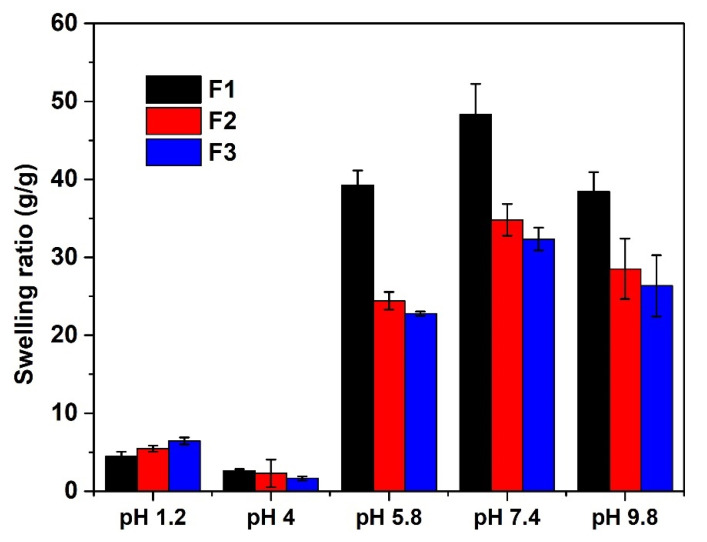
Swelling ratio values of CS-P (AAm-AA) at different pH (with F1, F2 and F3 prepared with 100, 150 and 200 mg of *N*,*N* methylene bisacrylamide, respectively).

**Figure 6 polymers-13-03571-f006:**
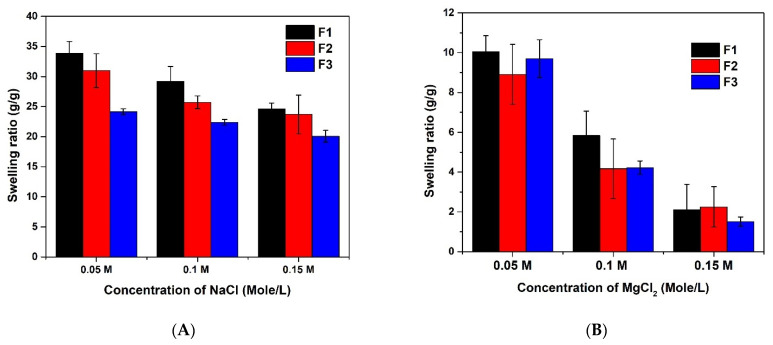
Effects of ionic strength on the swelling capacity of CS-P -(AAm-AA) hydrogel: (**A**) NaCl, (**B**) MgCl_2_, (**C**) AlCl_3_, and (**D**) comparison of NaCl, MgCl_2_ and AlCl_3_ (with F1, F2 and F3 prepared with 100, 150 and 200 mg of *N*, *N* methylene bisacrylamide, respectively).

**Figure 7 polymers-13-03571-f007:**
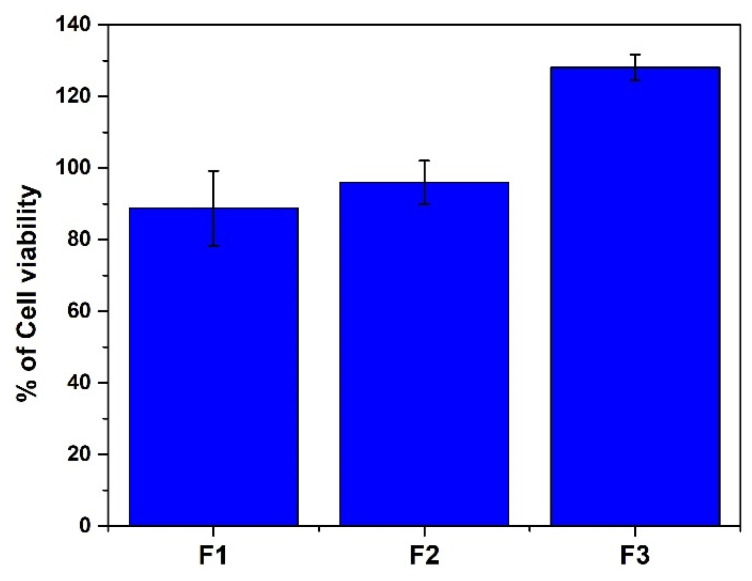
The cell viability % of HeLa cells tested with CS-P (AAm-AA) hydrogel at 50 µg/mL after 24-h incubation. Each point was the mean ± SD (standard deviation) of three independent experiments performed in triplicate (with F1, F2 and F3 prepared with 100 mg, 150 mg and 200 mg of *N*,*N* methylene bisacrylamide, respectively).

**Figure 8 polymers-13-03571-f008:**
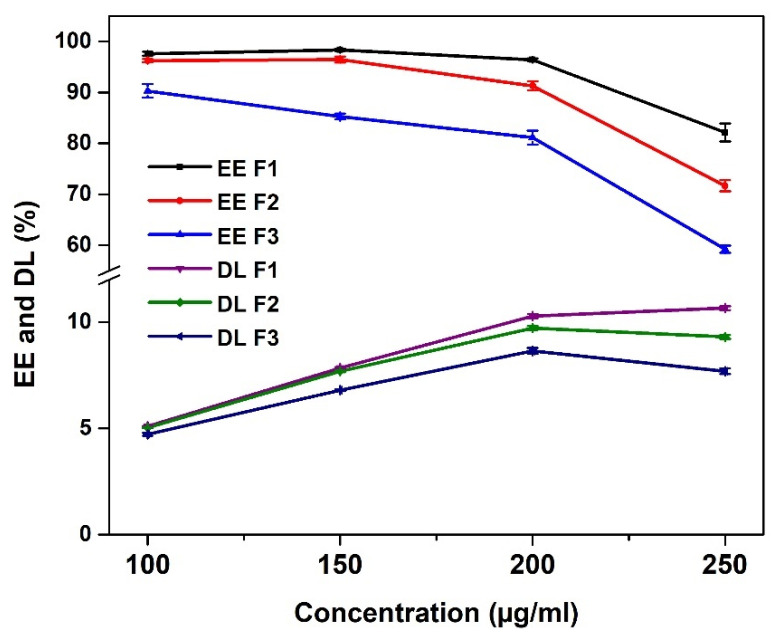
Optimization of encapsulation efficiency and drug loading (with F1, F2 and F3 prepared with 100, 150 and 200 mg of *N*,*N* methylene bisacrylamide, respectively).

**Figure 9 polymers-13-03571-f009:**
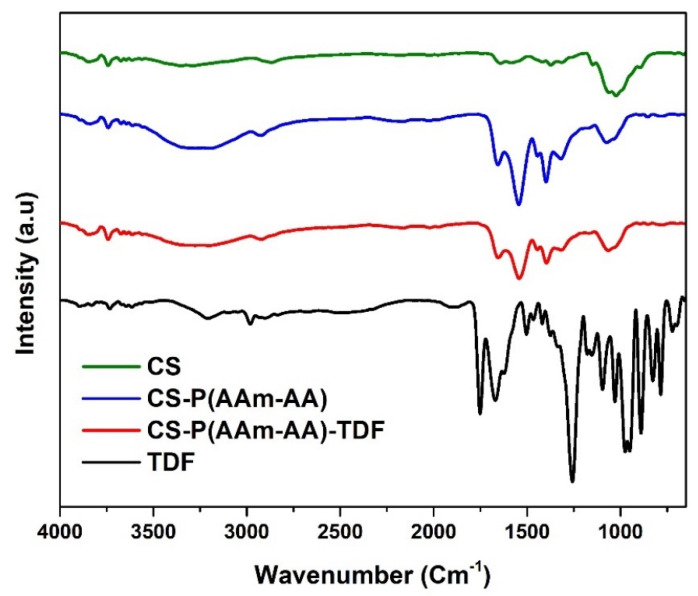
FTIR Spectrum of CS, CS-P (AAm-AA), CS-P (AAm-AA)-TDF and TDF.

**Figure 10 polymers-13-03571-f010:**
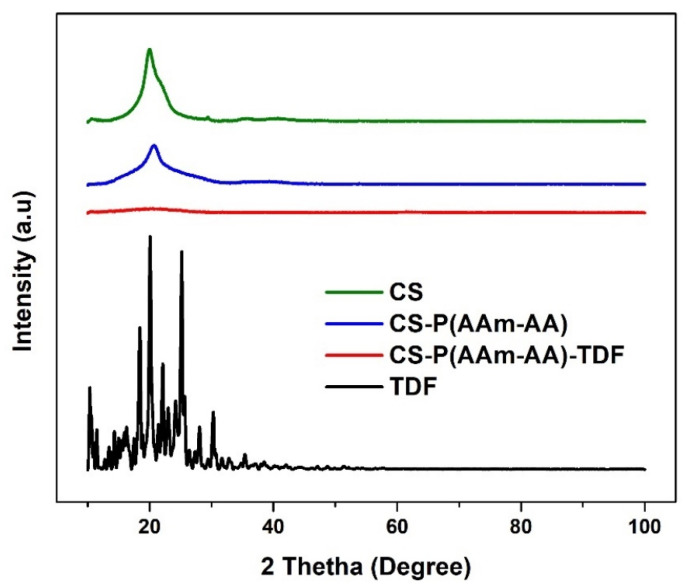
XRD Spectrum of CS, CS-P (AAm-AA), CS-P (AAm-AA)-TDF and TDF.

**Figure 11 polymers-13-03571-f011:**
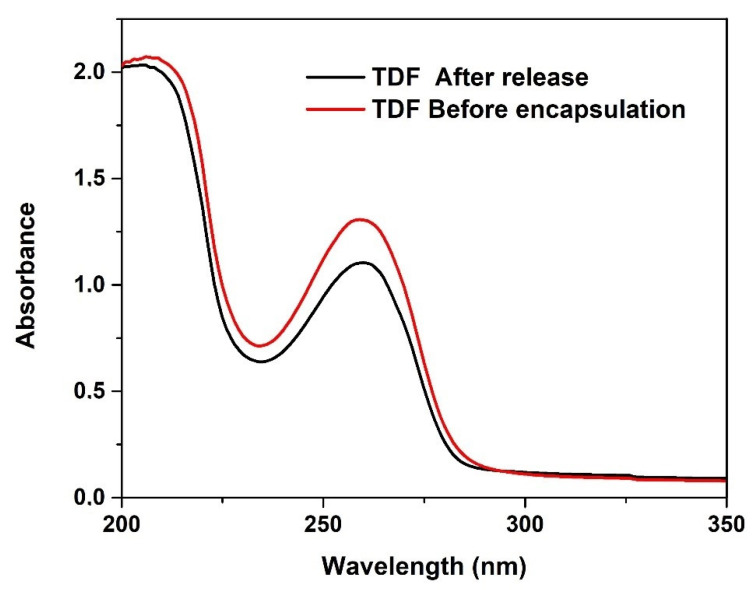
UV-Vis spectrum of pure TDF and TDF from in vitro release.

**Figure 12 polymers-13-03571-f012:**
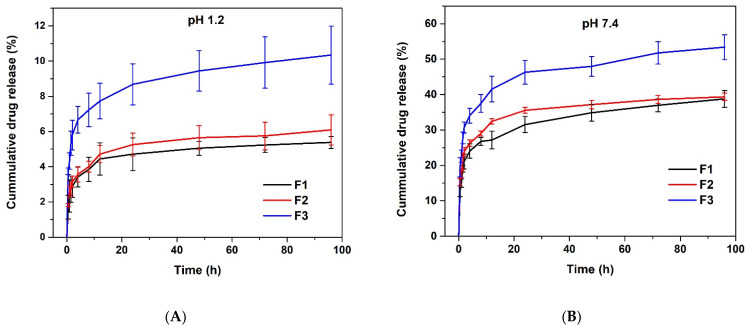
In vitro release of TDF (**A**) at pH 1.2 and (**B**) at pH 7.4.

**Table 1 polymers-13-03571-t001:** Composition of hydrogels formulation.

Formulation	Chitosan (mg)	Acrylamide (mg)	Acrylic Acid (mg)	*N*,*N* Methylene Bisacrylamide (mg)	Ammonium Persulfate (mg)
F1	500	1500	500	100	100
F2	500	1500	500	150	100
F3	500	1500	500	200	100

## Data Availability

The data presented in this study are available on request from the corresponding authors.

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
