# Peer review of "Development of pH-Sensitive Chitosan-g-poly(acrylamide-co-acrylic acid) Hydrogel for Controlled Drug Delivery of Tenofovir Disoproxil Fumarate"

_polymers, 2021, doi:10.3390/polym13203571_

Round 1

Reviewer 1 Report

The following corrections should be made:

  • In line 102: hence the ratio chitosan/syn- 102 thetic monomer was 0.25. Following monomer addition, variable amounts of N,N meth- 103 ylene bisacrylamide 100-200 mg (5-10 ml of 2% MBA solution) (Table 1) were added and 104 the reaction mixture was continuously stirred for 60 minutes. Please indicate temperature¡¡
  • In line 154 : In a dialysis bag, dried TDF-loaded hydrogel was im- 154 mersed in 25 mL of buffer solutions (pH = 1.2 and 7.4, I = 0.1M). Indicate which dialysis bag is used in methods?
  • In line 279, 295: Figure 5. Swelling ratio values of CS-P(AAm-AA) at different pH (with F1, F2 and F3 pepared with 279 100, 150 and 200 mg of N,N methylene bisacrylamide, respectively). Correct spelling of prepared, hydrogel…, N,N-methylene 296 bisacrylamide,

Author Response

Dear Reviewer,

Reviewer 2 Report

In this submission, the author has synthesized pH-sensitive chitosan-based chemically crosslinked hydrogel by free radical polymerization to graft acrylamide and acrylic acid using N,N’-methylene bisacrylamide as a crosslinker. The pH-dependent swelling mediated controlled delivery of an anti-hepatitis B drug, tenofovir disoproxil fumarate (TDF) was reported for over 100 hours.

The manuscript has been well written, except a few minor issues. The manuscript could be published after considering the comments and suggested modifications.

Here are my comments which need to be addressed.

  1. TDF is an anti-hepatitis B drug and HeLa is a cervical cancer cell line. It would be good to have rationality for choosing HeLa cell for determining cell viability.
  2. In the introduction part it is beneficial to include the information that hydrogel has Hamaker constant similar to water which is one of the reasons for its higher aqueous dispersion than other forms of drug delivery materials.

Author could find this information along with stimuli-responsive nano and micro-sized hydrogels as interesting papers to include in the introduction part. These are………….

  • https://pubs.acs.org/doi/pdf/10.1021/acs.chemmater.6b03440
  • https://pubs.acs.org/doi/pdf/10.1021/acsami.0c03689
  1. What is the degree of de-acetylation for pristine chitosan?

Did the author observe any decrease in the degree of de-acetylation during heating at 100C with NaOH for 1h (Fig 1)?

  1. In Figure 3, It’s needed to be mentioned the exothermic or endothermic direction on the y-axis with the legend, heat Flow.

  1. It would be good to have an explanation for why F1 group has low cell viability than F3 and followed an increasing trend from F1 to F3 (Fig. 7).

  1. Please include the legend on the Y-axis of Figure 9, as Intensity (a.u.)

Also, the same comment for Figure 10.

Please correct the spelling of 'degree' on the X-axis of figure 10.

Author Response

Dear reviewer,
